# The Nature and Cost of Readmissions after Work-Related Traumatic Spinal Injuries in New South Wales, Australia

**DOI:** 10.3390/ijerph16091509

**Published:** 2019-04-29

**Authors:** Lisa N. Sharwood, Holger Möller, Jesse T. Young, Bharat Vaikuntam, Rebecca Q. Ivers, Tim Driscoll, James W. Middleton

**Affiliations:** 1Faculty of Medicine and Health, University of Sydney, Sydney 2006, Australia; bvai6198@sydney.edu.au (B.V.); james.middleton@sydney.edu.au (J.W.M.); 2Kolling Institute, John Walsh Centre for Rehabilitation Studies, Sydney 2065, Australia; 3The George Institute for Global Health, University of New South Wales, Sydney 2042, Australia; hmoeller@georgeinstitute.org.au (H.M.); rebecca.ivers@unsw.edu.au (R.Q.I.); 4Justice Health Unit, Centre for Health Equity, Melbourne School of Population and Global Health, The University of Melbourne, Victoria 3010, Australia; jesse.young@unimelb.edu.au; 5Centre for Adolescent Health, Murdoch Children’s Research Institute, Victoria 3052, Australia; 6School of Population and Global Health, The University of Western Australia, Perth 6009, Western Australia, Australia; 7National Drug Research Institute, Curtin University, Perth 6102, Western Australia, Australia; 8School of Public Health and Community Medicine, University of New South Wales, Sydney 2052, Australia; 9School of Public Health, University of Sydney, Sydney 2006, Australia; tim.driscoll@sydney.edu.au; 10Agency for Clinical Innovation, Sydney 2067, Australia

**Keywords:** work-related injuries, spinal trauma, record linkage data, cost, rehabilitation, complications, unmet needs, unplanned readmissions

## Abstract

This study aimed to measure the subsequent health and health service cost burden of a cohort of workers hospitalised after sustaining work-related traumatic spinal injuries (TSI) across New South Wales, Australia. A record-linkage study (June 2013–June 2016) of hospitalised cases of work-related spinal injury (ICD10-AM code U73.0 or workers compensation) was conducted. Of the 824 individuals injured during this time, 740 had sufficient follow-up data to analyse readmissions ≤90 days post-acute hospital discharge. Individuals with TSI were predominantly male (86.2%), mean age 46.6 years. Around 8% (*n* = 61) experienced 119 unplanned readmission episodes within 28 days from discharge, over half with the primary diagnosis being for care involving rehabilitation. Other readmissions involved device complications/infections (7.5%), genitourinary or respiratory infections (10%) or mental health needs (4.3%). The mean ± SD readmission cost was $6946 ± $14,532 per patient. Unplanned readmissions shortly post-discharge for TSI indicate unresolved issues within acute-care, or poor support services organisation in discharge planning. This study offers evidence of unmet needs after acute TSI and can assist trauma care-coordinators’ comprehensive assessments of these patients prior to discharge. Improved quantification of the ongoing personal and health service after work-related injury is a vital part of the information needed to improve recovery after major work-related trauma.

## 1. Introduction

The immediate period following hospital discharge after acute traumatic injury holds substantial risk for ongoing health, recovery and welfare concerns [1]. Ongoing challenges can include physical limitations, continuing pain and the experience of secondary conditions. These factors can impair participation in previous activities, complicate or hamper the return to work [2] and contribute to financial stress, relationship strain and social exclusion. These factors predispose individuals to deterioration in mental health [3,4]; further impeding recovery from injury. Direct and measurable costs to an injured worker incorporate the loss or change in employment earnings, in addition to legal and medical costs. Indirect and less easily measured costs include pain, impaired function, reduced quality of life, potential for chronic injury and stress on interpersonal relationships [5,6]. Harmful substance use has previously been reported as high in injured populations, and likely compounds this multitude of issues [7,8]. Context-specific costs to the healthcare service for readmissions after acute injury are less well described, yet important to measure. 

The incidence and cost of traumatic spinal injuries (TSI) sustained in workplaces in New South Wales (NSW) was recently estimated for the first time. Over a 3-year period, 824 persons sustained TSI in work-related incidents, occupying 13,302 acute care bed days and costing a total of $19,500,000 (95%CI $16 M–$23 M) [9]. The total cost of work-related injury and disease in Australia was estimated at $61.8 billion in 2012–2013, of which NSW bore 28% ($17.3 billion) of the total cost from 31% of cases nationwide [10]. Injuries sustained in this cohort comprised column fractures, spinal cord injury or both; 21% of persons also sustained concomitant head, chest or abdominal traumatic injury. High numbers of these injuries occurred in the construction industry, particularly due to falling from height. This study did not report on any incidence, nature and cost of readmissions to hospital following the acute care period. 

Gabbe and Nunn recently reported 40% of patients after traumatic spinal cord injury to experience readmissions within the first 2 years in Victoria [11]. Similar proportions were described by Ruseckaite et al. [12] studying a cohort of patients injured in compensable transport incidents. Over one third of these patients experienced acute care facility readmission within 28 days of injury. Investigating unplanned readmissions in certain injury populations should be routinely undertaken to benchmark discharge planning efforts and therefore assist trauma care coordination.

Unplanned hospital readmissions occurrences within 28 days post the acute-care episode are progressively being used in various jurisdictions internationally as a measure of the quality of hospital care or treatment. In some systems, rates of unplanned readmissions are being used as an indicator of hospital performance that is then linked with funding reimbursements. Previous analysis in NSW has indicated that at least one quarter of unplanned readmissions are linked to “deficiencies in care” [13]. In 2013–2014, the NSW average for unplanned readmissions across all conditions and acute care facilities was 6.8 per cent; higher than the NSW 2021 target of 5.5 per cent [13]. 

The burden of TSI occurring in workplace settings is of particular interest for national injury prevention bodies, such as Safe Work Australia, who have prioritised a national target of a 30% reduction in serious work-related injury compensation claims by 2022 [14]. Quantification of the extended burden of TSI within the acute care setting complements the evidence informing injury prevention efforts in occupational health and safety, describing more specifically the extent of enduring disability associated with work-related injury. 

Therefore, the aim of this population-based study was to measure the post-acute care burden in a cohort of hospitalised traumatic spinal column and cord injuries that occurred while “working for an income” in NSW, Australia. Specifically, following those patients previously identified as sustaining a work-related TSI [9], we describe the incidence of 28- and 90-day readmissions, subsequent to their discharge from acute hospitalisation, investigating readmission etiology and healthcare system costs.

## 2. Methods

The epidemiology and occupational context of persons who sustained a TSI while “working for an income” in NSW has been previously and fully reported [9]. The Centre for Health Record Linkage (CHeReL) linked NSW Admitted Patient Data Collection (APDC), Registry of Births, Deaths and Marriage and Activity Based Funding (ABF) costing records for all people aged ≥16 years who were hospitalised between 1 June 2013 and 30 June 2016 with a TSI recorded in their index admission due to a work-related incident defined as ICD10-AM [15] code U73.0 or funding by workers compensation in the index admissions. Spinal injuries included all spinal cord and/or column injuries, defined using specific ICD-10-AM codes (Appendix A). NSW is the most populous Australian state with approximately 7.5 million inhabitants residing across 800,000 km^2^ in suburban, rural and very remote areas [16]. 

For the current study, readmissions were identified from APDC records linked with these index admissions. We defined hospital readmissions as being within 28 and 90 days after discharge with a primary diagnosis code related to the index admission. This benchmark was chosen in accordance with definitions from the NSW Bureau of Health Information [17], and the Independent Price Health Authority [18], who determine that readmissions to hospital within 28 days of discharge should be reviewed and considered as “potentially avoidable”. These were identified by manual review of the primary diagnosis codes of all admissions within 28 and 90 days, respectively. Readmissions within the first 28 days were consequently nested within those identified within 90 days.

Estimation of the costs of readmission was based on the formula used to calculate the National Weighted Activity Units (NWAUs) assigned to each ABF activity [19]. Weighted Activity Units (WAU) are the weights assigned per hospital separation; we used the 2013/2014 National Pricing Model technical specifications published by the Independent Hospital Pricing Authority [19] (for compatibility with AR-DRG version 6.0x). The WAUs in this study were adjusted for private patient service and private patient accommodation adjustments; a slight difference to NWAUs to avoid distortion by differential funding of public and compensable patients. All patients were assumed to be funded by Medicare for the WAU estimation. The “per separation” cost was defined as the WAU multiplied by the National Efficient Price 2013/2014 ($4993); with higher WAUs thus being more resource intensive.

Descriptive statistics were used to report the prevalence of various factors. Values were reported as mean and standard deviation (SD) for normally distributed continuous variables, proportions for categorical variables and median and interquartile range (IQR) for non-normally distributed continuous variables. All statistical analyses were performed using Stata version 15.0 (Stata Corporation, College Station, TX, USA). Standardised reporting of demographic and other variables, as recommended by De Vivo et al. [20], was followed where possible. 

This study was approved by the Cancer Institute NSW, Population & Health Services Research Ethics Committee: AU RED Reference: HREC/16/CIPHS/19, Cancer Institute NSW reference number: 2016/07/647.

## 3. Results

Between 1 June 2013 and 30 June 2016, 824 individuals sustained a traumatic spinal injury in NSW while working for an income, of which 740 (89.8%) had sufficient follow-up data to analyse readmissions <90 days post-acute hospital discharge (up to 1 April 2016) and were included in the analyses. Of these 740 individuals, 61 (8.2%) experienced a total of 119 (16.1%) readmissions in the first 28 days after their acute care discharge. By 90 days after discharge from acute care, 102 (13.8%) individuals had experienced 250 readmissions.

Characteristics of the primary injury cohort compared with individuals requiring readmissions within 28 and 90 days are described in Table 1. The cohort were predominantly male (86.2%); those aged 45–59 years were the largest age group (34.7%), this was also true for the readmitted proportions (37.7% and 36.3% for 28- and 90-day readmissions, respectively). Almost half of original work-related injuries had occurred as a result of a fall in their workplace (49.2%); over half of these (55.1%) were falls from building structures, scaffolding or ladders.

Table 2 displays the principal diagnosis (i.e., primary reason) for the readmission after the index TSI hospitalisation within 28 and 90 days; hence the denominator is the number of admission episodes, not the number of persons. Allied Health Services collectively apportion the majority of primary reasons for admission (56.3% and 59.6%). While this is a broad-reaching category, most individuals requiring readmission in this category were coded as “Care involving use of rehabilitation procedure, unspecified” (88%). All of these individuals had previously been discharged from an acute-care admission. Greater than 10% of the cohort of persons with TSI required readmission for complications of internal fixation devices (including infection), open wound infections and other complaints; 16 persons (13.4%) were readmitted within 28 days and 35 (14%) within 90 days.

The measure of resources required to accommodate these readmissions was counted as bed days and costs. The mean ± SD length of stay for readmissions within 28 days was 3.1 ± 7.0 days, compared with the length of stay for readmissions within 90 days at 4.8 ± 20.3 days. The total number of acute care bed days used by the 102 persons readmitted in 250 episodes was 1232 days at a total cost of $708,464 (95%CI: $417,325–$999,603). The mean ± SD per patient readmission cost was $6946 ± $14,532. The mean costs for 90-day readmissions were highest for patients with spinal cord injuries ($24,558), persons aged 16–29 years ($21,947) and transport incidents occurring on a street/highway ($15,492).

## 4. Discussion

In a cohort of 740 patients who had sustained work-related incident related traumatic spinal injuries during a three-year period, we found that around 8% (*n* = 61) of these patients experienced 119 inpatient readmissions within 28 days post-acute care discharge. Reasons for unplanned readmissions included requiring allied health interventions (56.3%), injury repair and operative complications (13.4%), circulatory problems including embolus (5%) and mental health problems (4.2%). The mean (SD) per patient cost for acute readmission was $6,946 ($14,532). Total costs of unplanned readmissions in the 90 days post-acute discharge were $708,464 (95%CI: $417,325–$999,602).

Although the recently reported NSW average of unplanned readmissions was 6.8% [13], we found a (disease-specific) readmission rate of 16.1% by 28 days post the index discharge among people who sustained work-related TSI, resulting predominantly from spinal fractures. Hospital performance monitoring in Australia measures a range of indicators including the incidence of hospital-acquired complications and rates of unplanned readmissions. These indices are now linked to government funding; for example, admissions where a hospital acquired complication incurs a payment reduction to hospital reimbursement [18]. Unplanned hospital readmissions can be a signal of issues with the effectiveness, continuity and integration of care provided to patients. As such, the use of patient clinical data in this way can help to drive safety and quality improvement. In this study, it may be that patients discharged to home may have rather benefited from a subsequent rehabilitation admission, avoiding the need to return to an acute care hospital. While rehabilitation admissions still impose a system cost, it is unlikely to be to the same extent. This information is therefore helpful in the discharge planning stage for patients with a traumatic spinal injury.

Our study had some limitations. We did not have access to various important variables about the injured worker, such as ethnicity, level of education and experience, the employment situation (e.g., whether permanent/part-time/casual) and specific occupation. Indigenous status was available but not identified from the APDC collection for this study. We have assumed all the patients to be of non-indigenous status for the estimation of costs using the NWAU-based approach; while we have not provided estimate of the cost impact this may have had, it is not anticipated to vary the current estimates significantly. Furthermore, the APDC District Network Return (DNR) does not include hospitalisation cost data for patients who were admitted to private hospitals across NSW. The degree of under-representation that this presents is uncertain, however such severe injuries are much more likely to be treated within the public hospital system [21]. We used NWAU-based costing approach to estimate the costs over the DNR data-based estimation to include all the public hospital separations in the costing analysis. The cost estimates presented are an under-estimation of the true costs as the separations at private hospitals were excluded from the analysis.

Braaf et al. [22] described the experiences of seriously injured patients within the Victorian state trauma service regarding communication from the health professionals caring for them after their injury. Many patients reported insufficient information, confusion and lack of clear communication to have hampered their discharge from acute care. While we did not interview patients involved in the current study, the high rate of readmissions for care involving rehabilitation suggests an area for improvement in the discharge planning process. Readmission can reflect the underuse of recommended care, adverse events and complications of hospital care, inadequate discharge planning or problems with coordination and integration of care across hospital, primary care and community settings. Patients discharged without sufficient information for continuity of care [23] can deteriorate once out of the acute care setting. Readmissions with infections, open wounds and device complications could likely be avoided with education, support and a general practitioner liaison.

## 5. Conclusions

Work-related traumatic spinal injuries create a significant burden of cost and disability for the Australian workforce but are preventable and also fall under a current focus of the Safe Work Australia policy to reduce serious injury compensation claims by 30% by 2022. This study demonstrates that the ongoing burden of work-related spinal trauma is not insignificant in the acute period post primary discharge. This study offers evidence of unmet needs after acute TSI and can assist trauma care coordinators’ comprehensive assessments of these patients prior to discharge. Improved quantification of the ongoing personal and health service after work-related injury is a vital part of the information needed to improve recovery after major work-related trauma.

## Figures and Tables

**Table 1 ijerph-16-01509-t001:** Characteristics of the injured cohort compared with readmitted individuals.

Characteristics	Cohort (*n* = 740) *N* (%)	Readmitted <28 days (*n* = 61) *N* (%)	Readmitted <90 days (*n* = 102) *N* (%)
Sex (male)	638 (86.2)	50 (81.9)	87 (85.3)
Age category years			
16–29	125 (16.9)	7 (11.5)	10 (9.8)
30–44	206 (27.8)	15 (24.6)	28 (27.4)
45–59	257 (34.7)	23 (37.7)	37 (36.3)
60+	152 (20.5)	16 (26.2)	27 (26.4)
Mechanism of injury			
Falls	364 (49.2)	29 (47.5)	48 (47.1)
Transport	223 (30.1)	16 (26.2)	29 (28.4)
Mechanical forces	124 (16.7)	11 (18.0)	18 (17.6)
Other/unspecified	27 (3.6)	5 (8.2)	7 (6.9)
Index Injury ^#^			
Spinal cord injury	44 (5.9)	7 (11.5)	13 (12.7)
Spinal fracture/dislocation	725 (97.9)	60 (98)	101 (99)
Cervical	161 (21.7)	11 (18.0)	20 (19.6)
Thoracic	258 (34.8)	25 (40.9)	36 (35.3)
Lumbosacral	424 (57.3)	35 (57.4)	65 (63.7)
Co-morbid traumatic brain injury	51 (6.9)	8 (13.1)	9 (8.8)
Co-morbid severe chest injury	86 (11.6)	11 (18.0)	19 (18.6)
Co-morbid abdominal injury	27 (3.6)	5 (8.2)	9 (8.8)
Operative procedure index admission	293 (39.5)	19 (31.1)	42 (41.1)
Persons to readmissions ratio	NA	61:119	102:250

^#^ Counts injuries, not persons, i.e., some patients have more than one injury. NA: Not applicable

**Table 2 ijerph-16-01509-t002:** Readmissions within 28 and 90 days.

Reason for Readmission *	28 days *N* (%) ^#^	90 days *N* (%) ^#^
Allied Health Service (Z00–Z99)	67 (56.3)	149 (59.6)
Care involving use of rehabilitation procedure, unspecified (Z50.9)	59 (88.0)	99 (66.4)
Other	8 (11.9)	50 (33.5)
Injury and related (S00–T98)	16 (13.4)	35 (14.0)
Mechanical complication/infection of internal device	9 (56.2)	12 (34.3)
Open wound/infection	5 (31.2)	8 (22.8)
Other (suppressed)	2 (12.5)	15 (42.8)
Musculoskeletal & nervous systems, skin, sensory organs (G00–G99, M00–M99, L00–L99, H00–H95)	9 (7.6)	19 (7.6)
Circulatory system (I00–I99)	6 (5.0)	9 (3.6)
Mental & behavioural (F00–F99)	5 (4.2)	9 (3.6)
Digestive & genitourinary systems (K00–K93, N00–N99)	5 (4.2)	12 (4.8)
Respiratory system & infections (J00–J99, A00–B99)	5 (4.2)	5 (2.0)
Not elsewhere classified (R00–R99)	6 (5.0)	12 (4.8)
**Total Readmissions**	**119**	**250**

* Principal ICD10-AM diagnosis group in brackets (15). **^#^** Of total readmission numbers.

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
