# Peer review of "The Nature and Cost of Readmissions after Work-Related Traumatic Spinal Injuries in New South Wales, Australia"

_ijerph, 2019, doi:10.3390/ijerph16091509_

Round 1

Reviewer 1 Report

I believe that the current results may provide significant implications for the medical care services in NSW.  The following is my observation:

It is suggested to further elaborate the importance of this topic in NSW.

For the part of Results, some terms or sentences may need to be further clarified, such as page 3 line 123 and 124, and page 4 line 143 and 144 (i.e. 3.1土7.0 days).  I think the presentation of current findings should be simple and clear.

In the Table 2, some specific codes had been shown, but it may not easy to be understand by other readers or audience.  Further elaborations are needed.

For the part of Discussion, I guess the authors can further discuss potential implications of current findings.  For example, how do we interpret the cost shown in the discussion? Potential reasons for supporting the readmission of the participants?  I think some aspects can be further elaborated.

Any suggestions can be provided according to the results or related discussions?

 I hope the above observation is useful, and thanks for giving me this opportunities to review this manuscript.

Author Response

Addressing Reviewer 1 Comments:

1.     It is suggested to further elaborate the importance of this topic in NSW.

We thank the reviewer for this opportunity to expand on spinal injury burden in the New South Wales (NSW) context in our manuscript. The economic burden of workplace-related spinal injuries in NSW has been further elaborated upon in our introduction (page 2, line 55 – 57).

2.     For the part of Results, some terms or sentences may need to be further clarified, such as page 3 line 123 and 124, and page 4 line 143 and 144 (i.e. 3.17.0 days).  I think the presentation of current findings should be simple and clear.

According to the reviewer’s suggestion, we have made minor changes in our presentation of our results to improve clarity and interpretability.

3.     In the Table 2, some specific codes had been shown, but it may not easy to be understand by other readers or audience.  Further elaborations are needed.

Although we appreciate the reviewer’s request for a concise presentation of results, our intent in including the ICD-10-AM diagnosis groups in table 2 was to facilitate future replication of study methodology and transparency. To aid readers who are not familiar with the ICD-10-AM classification system, we have added a reference to the footnote of the table to direct them to source the appropriate information to correctly interpret these codes. We have carefully reviewed Table 2 and believe that the information provided is essential information to understand the result presented in this table. Thus, we have decided not to change the overall presentation of Table 2.

4.     For the part of Discussion, I guess the authors can further discuss potential implications of current findings.  For example, how do we interpret the cost shown in the discussion? Potential reasons for supporting the readmission of the participants?  I think some aspects can be further elaborated.

We thank the reviewers for this opportunity to expand the discussion around the implication of these findings, indeed an important ‘take away’ message for translation. We have added text to the body of the Discussion (page 5, lines 166-176), and hope this speaks sufficiently to this point. 

5.     Any suggestions can be provided according to the results or related discussions?

We thank the reviewers for this opportunity and feel the insertion into the Discussion section, highlighted in response to the above question, should address this need.

Sincerely,

Dr Lisa N. Sharwood

Reviewer 2 Report

Thank you for the opportunity to review the manuscript titled “The nature and cost of readmissions after work-related traumatic spinal injury in NSW”. The manuscript is well-written and generally well presented; however, I have a few comments and suggestions.

In the title, ‘NSW’ should probably be fully spelled out. Also, given the international focus of the journal, consider adding ‘, Australia’ after ‘New South Wales’.

The authors seem to indicate that they used APDC data from public hospitals only (page 2, lines 86-87). Why only use data from public hospitals? I understand that vast majority of traumatic spinal injuries are likely to receive acute care at a public hospital, and that it is therefore unlikely that any index cases have been missed. However, some of the patients that receive acute care in a public hospital may continue to receive follow-up care (e.g. rehab) in a private hospital. Thus, if the present study did not include any data from private hospitals, the costs associated with readmission may be significantly underestimated.

Who conducted the data linkage? The CHeReL? This should probably be stated.

From the description in the Methods section, it is unclear whether this study is looking all readmissions or only unplanned readmissions. The subsequent Result and Discussion sections occasionally refer to unplanned readmission, albeit not consistently, thereby adding to the confusion. Please add a brief description of the operational definition for unplanned readmission, including how this was identified from the data.

The authors state that of the 824 cases “740 (89.9%) had sufficient follow-up data to analyse readmissions <90 days post-acute hospital discharge…” (page 3, lines 120-121). What does ‘sufficient follow-up data’ mean? Do the authors mean that they only have data up until the end of the study period (i.e. 30 June 2016)? And that they are therefore unable to follow up patients injured after 31 March 2016 for the full 90 days? Or are there any other reasons for ‘insufficient follow-up data’?

The authors state that the majority of unplanned readmissions were for rehab by allied health services (page 4, lines 135-138 & 154-155). Surely, such rehab would be planned, not unplanned? This goes back to my point above, how was an ‘unplanned readmission’ defined and identified?

The authors state that “indigenous status was not identified within the APDC” (page 5, lines 165-166). This is not true, the NSW APDC does in fact contain a variable for indigenous status; however, the completeness and accuracy of this variable is another matter.

The authors state that “the APDC does not include hospitalisation data for patients who were admitted to private hospitals across NSW” (page 5, lines 166-167). This is not true, the NSW APDC does in fact contain records of all inpatient separations from all public, private, psychiatric and repatriation hospitals in NSW.

What does ‘DNR’ mean (page 5, lines 170)? This abbreviation has not been previously explained.

In general, the Discussion section is somewhat weak. It contains very little ‘meat’ in terms of context and discussion that relates back to the issues highlighted in the Introduction section. What are the implications of this study?

Author Response

 Addressing Reviewer 2 Comments:

1.     In the title, ‘NSW’ should probably be fully spelled out. Also, given the international focus of the journal, consider adding ‘, Australia’ after ‘New South Wales’.

We appreciate the reviewer’s helpful suggestion. We have amended the title accordingly.

2.     The authors seem to indicate that they used APDC data from public hospitals only (page 2, lines 86-87). Why only use data from public hospitals? I understand that vast majority of traumatic spinal injuries are likely to receive acute care at a public hospital, and that it is therefore unlikely that any index cases have been missed. However, some of the patients that receive acute care in a public hospital may continue to receive follow-up care (e.g. rehab) in a private hospital. Thus, if the present study did not include any data from private hospitals, the costs associated with readmission may be significantly underestimated.

We thank the reviewer for pointing out this potential ambiguity. The record linkage included data from both the private and public hospitals, however the data from private hospitals was not included in the analysis as the cost data was unavailable for private hospital admissions. We have already acknowledged the current costs are an underestimate due to the exclusion of private hospital admissions (Page 7, line 173 – 174)

3.     Who conducted the data linkage? The CHeReL? This should probably be stated.

Data linkage was undertaken by the Centre for Health Record Linkage (CHeReL). We have amended the methods section (page 2, line 89) to make this clear for the reader. 

4.     From the description in the Methods section, it is unclear whether this study is looking all readmissions or only unplanned readmissions. The subsequent Result and Discussion sections occasionally refer to unplanned readmission, albeit not consistently, thereby adding to the confusion. Please add a brief description of the operational definition for unplanned readmission, including how this was identified from the data.

We thank the reviewer’s invitation to provide further clarification around this. Please see further clarification around this definition on p.3, lines 100-103. This should clarify that health departments are directed to consider 28 day readmissions as ‘potentially avoidable’. There is also some need for further clarification, noted in the IHPA reference, however on the whole, the 28 period readmission is predominantly considered to be not needing patients to return to hospital; a sufficient episode of care should have addressed needs – including referrals to rehabilitation if required.

5.     The authors state that of the 824 cases “740 (89.9%) had sufficient follow-up data to analyse readmissions <90 days post-acute hospital discharge…” (page 3, lines 120-121). What does ‘sufficient follow-up data’ mean? Do the authors mean that they only have data up until the end of the study period (i.e. 30 June 2016)? And that they are therefore unable to follow up patients injured after 31 March 2016 for the full 90 days? Or are there any other reasons for ‘insufficient follow-up data’?

We appreciate the opportunity to clarify our right censoring date for our study. Data was available until 30 June 2016, and hence only those patients injured until 01 April 2016 were included in the follow-up analysis. To clarify the duration of patient follow-up and resulting exclusions for the reader, we have added the right censoring date to the revised manuscript (page 3, line 124).

6.     The authors state that the majority of unplanned readmissions were for rehab by allied health services (page 4, lines 135-138 & 154-155). Surely, such rehab would be planned, not unplanned? This goes back to my point above, how was an ‘unplanned readmission’ defined and identified?

We thank the reviewer for identification of this point, and draw their attention to p.3, lines 100-103, where we have added a definition of (with pertinent reference) ‘unplanned readmission’.

7.     The authors state that “indigenous status was not identified within the APDC” (page 5, lines 165-166). This is not true, the NSW APDC does in fact contain a variable for indigenous status; however, the completeness and accuracy of this variable is another matter.

We thank the reviewer for pointing out this potentially confusing statement. The record linkage for this study did not include a measure of indigenous status and as such, we could not include Indigenous status in our analysis. We have assumed all the patients to be of non-indigenous status for the estimation of costs using the NWAU- based approach. Please see p.5, lines 180-5, for address to this.

8.     The authors state that “the APDC does not include hospitalisation data for patients who were admitted to private hospitals across NSW” (page 5, lines 166-167). This is not true, the NSW APDC does in fact contain records of all inpatient separations from all public, private, psychiatric and repatriation hospitals in NSW.

We appreciate the reviewer’s attention to detail. Although the APDC does include the hospitalisation data for private hospitals, the APDC DNR data does not have cost data for separations at private hospitals, and therefore we excluded those separations were excluded from the analysis. However, according to the reviewer’s suggestion, we have revised the manuscript to further clarify this for the reader (page 5, line 169-170).

9.     What does ‘DNR’ mean (page 5, lines 170)? This abbreviation has not been previously explained.

We apologise for this oversight. The DNR has now been expanded in page 5, line 169.

10.  In general, the Discussion section is somewhat weak. It contains very little ‘meat’ in terms of context and discussion that relates back to the issues highlighted in the Introduction section. What are the implications of this study?

We thank the reviewer for opportunity to expand on the implications of this study. We have provided additional context in the Discussion section, p. 5. Lines 166-176, and pages 199-201, to address this. We hope this responds to the reviewer’s request.

Sincerely,

Dr Lisa N. Sharwood

Round 2

Reviewer 1 Report

For this version of manuscript, it shows a significant improvement, and I do not have further comment on it.